# The Impact of Globalisation on the Development of International Fisheries Law

Kuan-Hsiung Wang [1] and Hung-Jeng Tsai [2,*]

1    Graduate Institute of Political Science, National Taiwan Normal University, Taipei 106, Taiwan; khwang@ntnu.edu.tw
2    Graduate Institute of Marine Affairs, National Sun Yat-sen University, Kaohsiung 804, Taiwan
*    Correspondence: hjtsai@mail.nsysu.edu.tw

**Abstract:** As the starting point, this paper introduces the development of globalisation and the evolution of the international legal order and then discusses the interaction between the two. The article then explores the impact of the "concept" and "practice" of globalisation on the evolution of the international fisheries' legal system by taking "sustainable development" and "international trade" as the probes to gain a practical understanding of the protection and conservation of high seas fisheries' resources. The authors argued that the international law of the sea is an ever-renewing legal system, especially in the regulation of conserving and managing high seas fisheries resources, which has undergone tremendous and drastic changes in recent decades due to the development of global trade, the strengthening of environmental issues, and the flourishing of international organisations. Obviously, globalisation is an important and fundamental driving force behind it. The authors presented findings by observing the changes in international fisheries law during the two decades between the signing of the UNCLOS and the completion of the IPOA-IUU.

**Keywords:** globalization; international fisheries law; international fisheries organisations; fish products; high seas fisheries





## 1. Introduction

If globalisation could be defined as the transnational flows of commodities, capital, labour and information, then the globalisation that began in the 1980s is the result of two major trends. The first is economic liberalization and the second is the advancement of information and communication technologies (ICT).

The world system we live in today is composed of two interconnected parts: the international political system dominated by individual nation-states and capitalism aimed at endless capital accumulation. The preservation of the sovereignty of nation-states (especially the ability to mobilize for war) requires the mobilization of domestic resources. Therefore, the efficient production capacity organized by capitalists constitutes one of the key supporting forces. On the other hand, the efficient operation of the market economy requires effective legal management and management of bureaucracy in every aspect, from the mobilization of production factors, production, sales, consumption, and financial management to the final accumulation of capital.

At the time of the gold standard, it was believed that there was a self-regulating market mechanism, free from political interference, that applied to both microscopic everyday transactions and the macroscopic global economy. However, the gold standard was actually operated on the basis of British hegemony. The First World War not only destroyed the gold standard, but also proved that states could effectively intervene in economic activities, such as production, distribution, and consumption, in order to mobilize for war. After the Second World War, the proposition that economic development requires effective government regulation and control, as elaborated by John M. Keynes, became the cornerstone of the Bretton Woods system.



However, after the stagnant inflation in the 1970s, Keynesianism was replaced by the supply side economics advocated by the UK and the US, which was based on the liberalization of trade, finance, and labour markets in order to increase productivity and eliminate inflation. This unveiled the introduction of neo-classic economics in the 1980s, which emphasizes the removal of government regulation and free flows of productive factors across state boundaries. Deregulation and transnational governance formed the premise of the World Trade Organisation (WTO) in 1995.

Institutional deregulation was reinforced by technological innovation. A crucial factor contributing to globalisation in the 1980s is the digitization of information and global network mobility created by ICT, a general technology that serves as the basis for other technological applications. At the same time, the accelerated flow of knowledge due to the advancement of information technology had in turn accelerated the development of innovative information technologies. The mutual reinforcement between the use of knowledge and the creation of knowledge, the so-called "feedback loop" of knowledge, rapidly developed into a transnational information network, which promoted not only global supply chains but also global governance constituted by formal intergovernmental negotiation and collaboration of non-government organisations [1].

The feedback loop effect between the use and creation of IT has changed the traditional vertically integrated industrial division of labour. Rapid technological innovation allows each production segment in the information industry to develop further technological differentiation, thus facilitating the transformation of the industry organisation from vertical integration to vertical differentiation. In the more differentiated technological levels, each country seeks its own production niche, and together they form a networked global production system that is more flexible in the organisation and more interdependent in function, also known as global supply chains. Global production is not limited by geographical boundaries, and can truly utilize the cheapest factors of production with comparative benefits in economics to carry out the most efficient production, and at the same time, accelerate the development of inter- and intra-industrial trade, of which computers, electronic products and automobiles are the most typical examples.

As a more elaborated global division of labour promotes increased productivity, it creates the crisis of resource exhaustion and environmental deterioration. In 1983 the United Nations established the World Commission on Environment and Development, which publicized "Our Common Future: Report of the World Commission on Environment and Development" in 1987 UN, *Report of the World Commission on Environment and Development: Our Common Future* (1987). [2]. The report pointed out that past economic growth patterns and governance, based on the sovereignty of individual nation-states, had caused the global ecological crisis to extend into interlocking crises ([2], para. 15).

These related changes have locked the global economy and global ecology together in new ways. In the past, we were concerned about the impacts of economic growth on the environment. We are now forced to concern ourselves with the impacts of ecological stress-degradation of soils, water regimes, atmosphere, and forests upon our economic prospects. More recently, we were forced to face up to a sharp increase in economic interdependence among nations. Ecology and economy are becoming ever more interwoven locally, regionally, nationally, and globally into a seamless net of causes and effects.

To deal with these interlocking crises, *Our Common Future* proposes that global development in the future must be sustainable development, whose definition is "to meet the needs of the present without compromising the ability of future generations to meet their own needs" ([2], para. 27). Sustainable development set the main tone of the Earth Summit at Rio de Janeiro in 1992, which formulated a series of institutional and legal reforms and funding sources required to establish an integrated management mechanism to solve global ecological degradation. Although these declarations and conference reports are not legally binding, they still provide the legal framework for normative perspectives to deal with global environmental problems. It is also under this transformation that the number of global NGOs rapidly grew and formed a bottom-up network of global governance.

The aforementioned observations are also reflected in the development of the international legal system. Since the adoption of the United Nations Convention on the Law of the Sea (UNCLOS) in 1982, the international fishery legal system has gone through a surging development and evolution. This reveals the impact of certain globalisation issues, such as "sustainable development and utilization of resources", "international trade and the environment", and "strengthening conservation and management measures of international fishery organisations" on the development of international fisheries law.

In the past four decades, since the signing of the UNCLOS, the evolution and development of international fisheries law have gained considerable impetus. This is particularly evident in the high seas fisheries, where some conditions have emerged for what was originally a matter of freedom of fishing on the high seas. For example, there is the competence of coastal states to manage fisheries resources, and there are limits to which attention should be paid in catching highly migratory fish stocks and straddling fish stocks. As a result, many factors and achievements can be seen that have been involved in the development of international fisheries law. In this paper, the authors present the findings by assessing changes in international fisheries law roughly during the two decades between the signing of UNCLOS and the completion of IPOA-IUU [3]. Other important developments, such as the EU IUU Regulation (Council Regulation No. 1005/2008 of 29 September 2008) and Agreement on Port State Measures to Prevent, Deter and Eliminate Illegal, Unreported and Unregulated Fishing, PSMA (2009) will be dealt with in another study.

## 2. The Impact of Globalisation on Sovereignty

The claiming and exercising of sovereignty is fundamental in international relations and international law because states, being the basic constituent units of the international community, have long advocated for the international law principle of "*par in parem non habet imperium*", and this principle is also manifested in the United Nations Declaration on Principles of International Law concerning Friendly Relations and Cooperation Among States in accordance with the Charter of the United Nations in 1970, in which it is said that all nations are entitled to sovereign equality, regardless of differences in their economic, social, political or other conditions. Sovereign equality includes, *inter alia*, the following elements:

(a)  States are judicially equal;
(b)  Each state enjoys the rights inherent in full sovereignty;
(c)  Each state has the duty to respect the personality of other states;
(d)  The territorial integrity and political independence of the state are inviolable;
(e)  Each state has the right to freely to choose and develop its political, social, economic and cultural systems;
(f)  Each state has the duty to comply fully and in good faith with its international obligations and to live in peace with other states [4].

Under the traditional consideration of international politics, the processing of international affairs is based on a state's practices in accordance with their respective tangible geographical boundaries as the scope of rights, supplemented by cooperation with international organisations or agreements with other states, in order to maintain or develop international relations. However, with the diversified development of globalisation issues, states are facing international affairs covering the aspects of politics, economy, law, technology, culture, media communication, environmental ecology, and social development. It is interesting to observe that the relationship between these aspects are overlapped and highly interactive, and it is difficult to differentiate them. Moreover, among those aspects, the economic factor is probably one of the most common and far-reaching elements for further consideration. It is not the only factor, but the development of economic globalisation has become the most important phenomenon among the various aspects of globalisation.

David Held analysed the development of globalisation and concluded that it includes four different issues: firstly, globalisation transcends political boundaries, as it is an extension of political and economic activities; secondly, the target of globalisation is the flow of trade, investment, finance, and culture; thirdly, globalisation will promote states' interac-

tions and accelerate global exchanges; and fourthly, globalisation will generate influence on those issues even if they have happened far away [5]

Therefore, the operation of national sovereignty in international law has undergone changes in state practices under the rapid development of globalisation. In the case of the above-mentioned discussion on the sovereign equality of states, agreement or consensus among the States is needed. However, the reality is that states perform slowly when they are dealing with transnational issues. Under such circumstances, international organisations have gradually played more important roles in resolving those issues. In the study of globalisation, this often refers to inter-governmental organisations (IGOs), non-governmental international organisations (NGOs), and multinational corporations (MNCs). Although all three are playing active roles in the development of international relations, this paper will take IGOs as the object of discussion, since the formation of the international legal order is based on the states as the main bodies.

It is understandable that there are similar aspects in the functions of international organisations and national sovereignty through long-term practices. International organisations have independent legal personalities [6]. Furthermore, international organisations are established by sovereign states through treaties. The existence of an international organisation requires the consent of or authorization from a sovereign state. Moreover, an international organisation is capable of enforcing its resolutions on member states, and even on non-member states.

Taking high seas fisheries as an example, fisheries refer not only to the fishing activities themselves but also the shipbuilding industry, personnel, etc., but also to recruitment, welfare, fishing gear, port equipment, management, transportation, fishery product production, trade and even marketing [7]. However, it is impossible for one country to maintain its fishing industry through domestic control, either from its own legislation or from its practices in fisheries. In contrast, the globalisation of fish, the fish products trade, the regulations for such practices, and even the governance of international fisheries are all manifested in concrete ways in fishing activities.

However, the emphasis on globalisation (such as international trade practices and the flourishing development of international organisations) conflicts with the traditional concept of respect for national sovereignty in international law. This is a result of the fact that the resolution of current global international issues can no longer be achieved only by a single state, or even when the State is willing to transfer part of its sovereignty to share interests with others. This undermines the basis on which national sovereignty is built. Therefore, the concept of traditional national sovereignty has been challenged in the globalised world, and some scholars have argued that sovereignty is gradually being weakened and replaced by another form of exercise [8]

### 3. The Impact of Globalisation on Fishing Activities

The United Nations Food and Agriculture Organisation (FAO) has repeatedly issued warnings about the state of marine living resources, pointing out that more than 60% of major fisheries are fully exploited or overexploited, while 35% are severely exploited. Facing the predicament of the possible overexploitation of fishery resources, the Consensus on World Fisheries was adopted at the ministerial meeting held by FAO in March 1995. The document clearly states that the international community needs to take a number of actions, such as eliminating overfishing, rebuilding and strengthening fish stocks, reducing wasteful fishing practices, and developing new and alternative fish stocks based on scientific sustainability and responsible management [9]. It also warns that if the aforementioned actions are not implemented, about 70% of the fish stocks on the planet will continue to decline, and these are fish stocks currently considered to be fully exploited, overexploited, depleted, or in the process of recovery ([9], para. 7).

In addition, according to the FAO *the State of World Fisheries and Aquaculture 2022*, [10], from 1976 to 2020, the value of trade in aquatic products increased at an average annual rate of 6.9% in nominal terms and 3.9% in real terms (adjusted for inflation). The faster

rate of growth in value relative to quantity reflects the increasing proportion of trade in high-value species and products undergoing processing or other forms of value addition. In terms of exports, China remains the world's largest exporter of aquatic animal products, followed by Norway and Vietnam, with the European Union as the largest single importing market. The largest importing countries are the United States of America, followed by China and Japan. In terms of volume (live weight), China is the top importing country of large quantities of species not only for domestic consumption but also as raw material to be processed in China and then re-exported.

It is understandable that fishery activities that were originally dominated by the state, no longer exist within the jurisdiction of the state. Instead, through the globalised trade in fishery products, transnational fishing activities, and the capture of transnational fish species, these transnational, trans-regional and even global activities have given a deeper meaning to fishery operations or activities, i.e., the international legal regime governing fishing operations at sea has been influenced by the environment protection consideration, the globalisation of trade, and the measures taken by international organisations.

### 3.1. Conservation and Management of Fishery Resources

As a result of the rapid pace of industrial modernisation and the globalisation of the sale of industrial products, the entire Earth has suffered from the after-effects of the development of civilisation. The deterioration of the ecological environment has led the international community to reconsider the protection of the global environment and the use of biological resources. States are aware of the fact that the global environment is indivisible, and if one ecosystem or one region is damaged, the effects will spread to other ecosystems or regions, and even to the whole world. For example, the destruction of the Antarctic ozone layer, global climate change, acid rain, desertification, reduction in tropical rainforests, transboundary transportation of hazardous waste, damage to the marine environment, and carbon dioxide emissions from various countries are all directly or indirectly harmful to the global ecological environment. Among these considerations, fishery activities are an extremely obvious area. The commercial transaction of fish and fish products and the active mobilization of fishing activities are worthy of observation because they may not only cause the depletion of living marine resources, but also the destruction of marine ecosystems.

If we only look at the phenomenon of marine environmental pollution and the over-exploitation of biological resources, overfishing and the deterioration of marine habitats are destroying the main source of human food. Rapid population growth, excessive land use, agriculture production, deforestation, fishery resources over-exploitation, urban development, and industrial emissions all affect the marine environment. According to the Sustainable Development Goals Report 2022, between 2009 and 2018, the world lost about 14% of its coral reefs, often called the "rainforests of the sea" because of the extraordinary biodiversity they support. The oceans are also under increasing stress from multiple sources of pollution, which is harmful to marine life and eventually makes its way into the food chain. Among those sources of pollution, 80% comes from land-based activities. In 2021, a study estimated that more than 17 million metric tons of plastic entered the world's oceans, making up about 85% of marine litter. The volume of plastic pollution entering the ocean each year is expected to double or triple by 2040, threatening all marine life. Moreover, in terms of fishery resources, global fish stocks are under increasing threat from overfishing and from IUU fishing. More than a third (35.4%) of global stocks were overfished in 2019, an increased compared to 34.2% in 2017 and 10% in 1974. In addition, the rapidly growing consumption of fish (an increase of 122% between 1990 and 2018), along with inadequate public policies for managing the sector, have led to depleting fish stocks [11,12].

The international community is aware that the pollution of the marine environment and the overexploitation of living marine resources are seriously undermining the productivity of the oceans and endangering the livelihoods of those who depend on them for their livelihoods. Through the signing and entry into force of the UNCLOS, the international

community sought to provide a comprehensive legal framework for the management of the oceans and seas. Part XII of the UNCLOS stipulates the sources of pollution from land, sea, and atmosphere, as well as the consequences of pollution resulting from economic development activities that exploit marine resources. Obviously, overfishing is one of those activities. However, this convention is just the beginning and there have been numerous meetings and documents on the regulation of high seas fishing activities.

### 3.1.1. The Emergence of the Concept of Responsible Fishing

The Mexican government convened the International Conference on Responsible Fishing in Cancun, Mexico, from 6 to 8 May 1992 The Cancun Declaration defines 'responsible fishing' as "meaning that the sustainable use of fisheries resources should be compatible with the environment; that fishing and farming activities should not harm ecosystems, resources or their quality; and that the valorisation of fish products or manufacturing processes should meet the requirements of health standards and provide products of good quality to consumers in the course of commerce" and that "subject to the relevant provisions of the UNCLOS freedom of fishing on the high seas should be balanced with the obligation to cooperate between states to ensure the conservation and rational management of biological resources" [13]. The conference further explained the concept of "responsible fishing" and discussed the current state of global fisheries, resources and environment, management and development, capture and trade of fishery products, and issued a Declaration of Cancun on the conservation and management of global fisheries resources. The Declaration requested FAO to consult with relevant international organisations to draft a Code of Conduct for Responsible Fishing, which takes into account the Declaration. However, it is a document expressing political willingness and it is not legally binding in its nature.

### 3.1.2. Chapter 17 of Agenda 21

The United Nations Conference on Environment and Development (UNCED) was held in Rio de Janeiro, Brazil from 3 to 14 June 1992. The conference discussed the issue of marine fisheries and concluded that all oceans should be protected, used wisely and exploited for their biological resources. The Rio Declaration on Environment and Development (Rio Declaration), [14], and Agenda 21 [15,16], were adopted, and Chapter 17 of Agenda 21 includes a section on the protection of the marine environment. Paragraph 17.46 of Chapter 17C "Sustainable use and conservation of marine living resources of the high seas" emphasizes [16] that states commit themselves to the conservation and sustainable use of marine living resources on the high seas. To this end, it is necessary to:

a. Develop and increase the potential of marine living resources to meet human nutritional needs, as well as social, economic, and development goals;
b. Maintain or restore populations of marine species at levels that can produce the maximum sustainable yield as qualified by relevant environmental and economic factors, taking into consideration relationships among species;
c. Promote the development and use of selective fishing gear and practices that minimize waste in the catch of target species and minimize by-catch of non-target species;
d. Ensure effective monitoring and enforcement with respect to fishing activities;
e. Protect and restore endangered marine species;
f. Preserve habitats and other ecologically sensitive areas;
g. Promote scientific research with respect to the marine living resources in the high seas.

Therefore, all states (whether coastal or distant-water fishing nations) have the responsibility to protect and manage the living resources in the high seas. As a result, the conservation and management of marine living resources have been widely debated by the international community and a consensus has been reached on the further conservation and management of high seas fisheries resources. Since the launch of Agenda 21, the international community has been actively discussing, formulating, and adopting a series

of documents for the conservation and management of marine resources with the goal of "sustainable development", although Agenda 21 itself does not have binding forces.

3.1.3. Regulation of Fishing Vessels Operating on the High Seas

At its 27th session in November 1993, FAO adopted the Agreement to Promote Compliance with International Conservation and Management Measures by Fishing Vessels on the High Seas (the Compliance Agreement) [17]. Its preamble begins by reaffirming the freedom of fishing on the high seas and the restrictions it faces when exercising such freedom. One of the main restrictions is that "all states have the duty to take, or to cooperate with other States in taking, such measures for their respective nationals as may be necessary for the conservation of the living resources of the high seas" ([17], paras. 1,2). The Compliance Agreement establishes and strengthens the means for the flag state to exercise jurisdiction and control over fishing vessels that have the right to fly its flag in accordance with the UNCLOS, and at the same time, promotes transparency in high seas fishing operations.

3.1.4. Regulation of Fishing Operations Targeting Straddling and Highly Migratory Fish Stocks

On 4 August 1995, the United Nations adopted the Agreement for the Implementation of the Provisions of the United Nations Convention on the Law of the Sea of 10 December 1982 relating to the Conservation and Management of Straddling Fish Stocks and Highly Migratory Fish Stocks (the UNFSA) [18]. The goal of the UNFSA is very clear; that is, to ensure the long-term conservation and sustainable use of straddling fish stocks and highly migratory fish stocks through the effective implementation of the relevant provisions of the UNCLOS ([18], Article 2). The UNFSA takes into account the integrity of marine ecology, so it is stipulated that the parties should also assess the impact of fishing, other human activities and environmental factors on the target population (target stocks) and species belonging to the same ecosystem or species related to or dependent on the target population. If necessary, conservation and management measures should be harmonized for species belonging to the same ecosystem or related species to maintain or restore the number of such species, so as to prevent the reproduction of species from being seriously threatened ([18], Article 5). At the same time, the UNFSA also clearly stipulates that coastal states and states fishing on the high seas shall have the duty to "cooperate" to adopt measures to ensure the long-term sustainability of straddling fish stocks and highly migratory fish stocks and promote the objective of their optimum utilization [18].

3.1.5. Establishment of the Code of Conduct for Responsible Fisheries

In October 1995, the resolution of the 28th session of the FAO adopted the Code of Conduct for Responsible Fisheries, [19]. It is a continuation of the 1992 Cancun Declaration. The purpose of the Code of Conduct is mainly to provide an international standard of conduct to promote the effective development, conservation, and management of fishery resources. That is to say, the principles of the Code of Conduct can promote the consistency between the development and utilization of fishery resources and the principle of "sustainable development". Moreover, it recognizes the important role of fisheries in world food security, economic and social development, and the need to ensure the sustainability of aquatic biological resources and their environment for present and future generations. Two points are worth noting. The first is that compared to the term of "fishing responsibility" used in the 1992 Cancun Declaration, the Code of Conduct uses "responsible fisheries". This change indicates an expansion of the definition of fisheries to include the whole range of fisheries' activities, rather than being limited to fishing alone; the second is that the Code of Conduct in its first article expresses that the document is voluntary in nature ([19], Article 1.1). It continues to say that certain parts of it are based on relevant rules of international law, including those reflected in the United Nations Convention on the Law of the Sea of 10 December 19821. The Code also contains provisions that may be or have

already been given binding effect by means of other obligatory legal instruments amongst the Parties, such as the Agreement to Promote Compliance with International Conservation and Management Measures by Fishing Vessels on the High Seas, 1993, which, according to FAO Conference resolution 15/93, paragraph 3, forms an integral part of the Code and it does not have a mandatory legal regulatory force. However, it appears from states practices that the Code of Conduct has become a fundamental instrument in the development of international fisheries law.

### 3.1.6. Other Instruments concerning Fishing Activities

The Kyoto Declaration and Plan of Action on the Sustainable Contribution of Fisheries to Food Security was adopted by delegates from 95 countries at a conference in Kyoto, Japan, from 4 to 9 December 1995. In the Kyoto Declaration [20], participants noted the trend of the increasing global population, the need to ensure a secure supply of food for present and future generations, the significant contribution of fisheries to income, wealth and food security, and the importance of fisheries to a number of low-income and undersupplied countries, and therefore, the importance of the concept of sustainable use of resources in order to promote the goal of maximizing the utilization of fish products.

The Rome Declaration on the Implementation of the Code of Conduct for Responsible Fisheries was adopted at the FAO Ministerial Conference held on 10–11 March 1999 [21]. In particular, Paragraph 4 of the Rome Declaration expresses that the participating states welcomed the adoption by the FAO Committee on Fisheries at its 23rd Session in February 1999 of International Plans of Action for the Management of Fishing Capacity (IPOA-Capacity), for the Conservation and Management of Sharks (IPOA-Sharks), and for Reducing Incidental Catch of Seabirds in Long-line Fisheries (IPOA-Seabirds) [22]. These International Plans of Action are all within the framework of the Code of Conduct for Responsible Fisheries. It is hoped that the content of these documents and the role of countries in policy formulation will lead to the development of national action plans to achieve the goal of sustainable use of high seas living resources.

In addition, in view of the increasing number of fishing activities on the high seas by fishing vessels of the flag of convenience and the need for the development of an international fisheries management system, the FAO Committee on Fisheries (COFI) adopted the International Plan of Action to Prevent, Deter and Eliminate Illegal, Unreported and Unregulated Fishing (IPOA-IUU) at its 24th session on 2 March 2001 [3]. It is expected that fishing vessels that are deemed as not regulated by international fisheries organisations do not comply with relevant conservation management measures and do not submit or submit false catch reports will be combated and eliminated. It is because such practices are contrary to the objectives of international fisheries management organisations to conserve fisheries' resources and have a significant negative impact on the sustainable use of fishery resources.

### 3.1.7. Johannesburg Global Sustainable Development Summit and Implementation Plan

Ten years after the aforementioned Rio Declaration in 1992, the World Summit on Sustainable Development (WSSD) in Johannesburg in 2002 adopted a Plan of Implementation [23], which set out a timetable for the sustainable development of fisheries resources:

By the end of 2004 → Preventing, deterring and eliminating illegal, unreported and unregulated fishing.
By the end of 2005 → Countries and regional fisheries organisations agree on effective, equitable and transparent management of fishing capacity on a global basis.
2006 → Substantial progress can be made to protect the marine environment from land-based activities.
By the end of 2012 → Establishment of representative networks and time/area restrictions on the protection of fish farms and periods.
By the end of 2015 → Maintain or restore populations to a level where they can produce maximum sustainable production.

While this approach demonstrates the international community's concern over the depletion of fisheries resources and the urgency of restoring them, the schedule for completion also demonstrates the binding effect that the WSSD Plan of Implementation seeks to achieve. However, as the situation stands today, the arranged schedule and outcome expectations appear overly optimistic.

Following the two Earth Summits and the progress of the concept of sustainable development of high seas fisheries resources, it is clear that the international community is increasingly concerned about the depletion of fisheries resources and the urgency of restoring them. Even in terms of drawing up a schedule, members of the international community have expressed great anxiety about the connection between the sustainable development of resources and the conservation of marine living resources.

*3.2. Fish Products Trade*

The scope of trade in fishery products is a standard global phenomenon, and through trade practices, the quality and quantity of capture of fishery resources is enlarging, and even the velocity of depletion of these resources is accelerating. As a result, there is a tendency for governments or international organisations to discuss environmental protection issues in conjunction with trade activities and to link them more closely. The following is an analysis of the relationship between trade and the environment in relation to the adoption of trade measures and the application of eco-labelling between countries.

Trade measures are used to influencing the formulation or modification of specific policies of other countries by adjusting trade activities. One of the famous examples is the one between the United States and Mexico over the conservation of dolphins, which can be traced back to the enactment of the US Marine Mammal Protection Act (MMPA) in 1972 to protect marine mammals from the harmful effects of human fishing activities. According to the Act, embargoes could be imposed on countries that did not meet US conservation and management standards.

In addition to this, the promotion of 'dolphin safe' eco-labelling has been established through private commercial activity in the USA. The concept of eco-labelling is to regulate fishing activities through commercial behaviour between the consumer and the producer. Through the use of the eco-labelling system, consumers are encouraged and educated to consume fish species that have been caught from sources that have been replenished. This approach is supported by companies, governments, and volunteer groups worldwide.

This dual approach of domestic legislation and domestic commercial pressure by the US government has put great pressure on many fishing nations around the world, one of which is Mexico. In 1990, the United States imposed an embargo on tuna produced in Mexico under the Marine Mammal Protection Act, but this was countered by Mexico under the terms of the General Agreement on Tariffs and Trade (GATT). The reason for this controversy is that dolphins and tuna often migrate together in the waters of the Eastern Pacific Ocean, which has led fishermen to use dolphins migrating at the surface as a reference target when fishing for tuna in deeper waters, but this led to by-catch. This practice has drawn the attention of several marine environmental and ecological conservation groups in the United States and has led to a boycott of Mexican fishing products by the US government.

The controversy over ecology and resource conservation has not diminished with the passage of time, or even with improved fishing laws in Mexico. In 1992, the Dolphin Conservation Program was established within the IATTC, the regional fisheries organisation for the Eastern Pacific Ocean region, to protect dolphins and reduce their mortality due to fishing operations. Under the programme, representatives from IATTC member governments, the fishing industry, and environmental conservation groups form the International Review Panel, which reviews reports from onboard observers, determined whether there had been a breach of operational regulations and recommends sanctions.

Statistics show the mortality rate of dolphins during fishing operations has decreased from 130,000 in 1986 to 3000 in 1994, which is a remarkable result [24]. Although the

US government lifted the embargo in 1997, a combination of boycotts by canners and distributors in the US led to restrictions on the sale of tuna products from countries such as Mexico, which have not yet been lifted and have led to trade conflicts between the US and Latin American countries, particularly Mexico. In addition to the case concerning the embargo on tuna products, there is also the case concerning the Import Prohibition of Certain Shrimp and Shrimp Products between Malaysia and the United States under the dispute settlement system of the World Trade Organisation [25]. In that case, the United States invoked its domestic law (Section 609 of the US Public Law 101-162) to prohibit the importation of shrimp products where the use of trawls to catch shrimp had caused the death of sea turtles. While this is certainly relevant to ecological conservation, it has prompted Malaysia to appeal to the World Trade Organisation to resolve the dispute. Meanwhile, Australia, the European Union, Hong Kong, China, India, Japan, Mexico, and Thailand have also submitted position papers for third-party countries.

Professor Richard Parker suggested that such trade sanctions can be abused in three ways by those who emphasize free trade: firstly, by protectionism in disguise; secondly, by imposing trade pressure specifically for environmental reasons; and thirdly, by imposing bilateral economic pressure on the sanctioning state to alter multilateral cooperation, or to impose excessive and unfair conditions on that cooperation [24].

Regardless of which assumption is consistent with future developments, the use of trade measures as a means of achieving environmental protection is already well established and should continue to be discussed in the context of future environmental issues. It is also worth noting that manipulating domestic legislation as a tool to influence the policies of other countries has become a common practice in non-traditional interactions between countries.

Another issue is an analysis of the application of eco-labelling. The so-called eco-labelling is a process of approving labels for fish products that have a lower environmental impact during the production process, with the aim of increasing the sustainable management of fisheries and educating consumers about environmental protection [26]. The concept of eco-labelling has already been discussed in the previous discussion of trade measures, with the aim of regulating fishing activities through commercial behaviour between consumers and producers. The European Union's view on eco-labelling is to provide guidance to consumers on products that aim to reduce environmental impacts during the biological life cycle and to provide information on the environmental characteristics of labelled products [27]. In general, in state practices, the use of eco-labelling encourages and educates consumers to consume fish or fish products that have been processed from sources that are sustainable.

Whether the use and requirements for eco-labelling will constitute a non-tariff (or technical) barrier to the trade of fish and fish products remains to be explored, but the impulsion of international non-governmental organisations is a demonstration of the impact of trade globalisation.

### 3.3. Efforts from Regional Fisheries Management Organisations (RFMOS)

In any case, by way of the integration capability, international organisations should have the effect of achieving governance on the conservation of fishery resources that are transnational in nature. This is evidence that resolutions adopted by international organisations have become, in substance, one of the sources of international law. [28] Therefore, Article 118 of the UNCLOS specifies how to conserve and manage high seas resources among countries. Furthermore, according to the provisions of the Convention on the Law of the Sea, with regard to the issue of straddling and highly migratory fish stocks, the second paragraph of Article 63 imposes this obligation on coastal states and fisheries that fish these stocks on the high seas states, they should agree or cooperate on ways to conserve the stock. Such cooperation can be achieved through bilateral or other agreements, as well as through appropriate sub-regional and regional organisations. In fact, Article 63, Paragraph 2 of UNCLOS has foreseen the importance of establishing a

cooperative mechanism for the conservation of fishery resources in high seas areas, which encourages cooperation through appropriate subregional or regional organizations [29]. In addition, Article 64 adds an additional obligation to coastal states and other high seas fishing states, expressly stating that such cooperation is to ensure the conservation of straddling and highly migratory fish stocks, with a view to improving the fishery resources inside and outside the exclusive economic zone, achieving optimal utilization. If no suitable international organisation exists to ensure such cooperation, Article 64 of UNCLOS stipulates that coastal states and other high seas fishing states fishing for these stocks "shall cooperate in the establishment of such organisations and participate in their work" [29]. The emphasis on the importance of this cooperative functioning mechanism is also clearly regulated in Agenda 21:

17.10. → International cooperation and coordination, on a bilateral basis and, where feasible, at a subregional, interregional, regional or global level, with the role of supporting and complementing the national efforts of coastal States to promote integrated management and sustainable development.

17.11. → States should cooperate, as appropriate, in the development of national guidelines for integrated coastal zone management and development, taking into account existing experience.

Following this design in UNCLOS, Part III of the 1995 Compliance Agreement emphasises international cooperation mechanisms, i.e., the establishment and functions of regional or sub-regional international fisheries organisations. More notably, in order to avoid impediments to the functioning of such an international fisheries organisation by non-parties, Article 17(1) of the 1995 UNFSA goes further by providing that:

A state that is not a member of a subregional or regional fisheries management organisation or is not a participant in a subregional or regional fisheries management arrangement, and which does not otherwise agree to apply the conservation and management measures established by such organisation or arrangement, is not discharged from the obligation to cooperate, in accordance with the Convention and this Agreement, in the conservation and management of the relevant straddling fish stocks and highly migratory fish stocks.

In international practice, there are various RFMOs that are organized on a geographic basis and have a long history of managing straddling and highly migratory fish stocks [30]. For example, the Western and Central Pacific Ocean Fisheries Commission (WCPFC) [31], the Inter-American Tropical Tuna Convention (IATTC) [32], the International Commission for the Conservation of Atlantic Tunas (ICCAT) [33], the Indian Ocean Tuna Commission (IOTC) [34]; and the Commission for the Conservation of Southern Bluefin Tuna (CCSBT) [35].

The resolutions and management measures adopted by these RFMOs have added regulative power on non-members or non-parties fishing in the waters under the RFMOs jurisdiction. For example, when the Antigua Convention enters into force, the Convention Area of IATTC was set to be the waters between the eastern boundary of the American continent's west coast, the northern and southern boundaries of 50° S and 50° N, and the western boundary of 150° W. The Convention Area of IATTC will also include the high seas. Then, there will be many discussions or even debates on such development concerning the connotation of public international law [36]. It is a well-known principle of "*pacta terrtiis nec nocent nec prosunt*", which means "a treaty binds the parties and only the parties; it does not create obligations for a third state". In this regard, this would be a reflection of the pressure on national sovereignty to compromise on facing the dilemma between the need for globalisation and the pursuit of national interests.

## 4. Conclusions

The development of globalisation has become an important phenomenon and has resulted in the development of the modern international society. It presents multiple aspects, including political, economic, cultural, and other aspects, within which, "economic globalisation" is a crucial part of globalisation, which means that various economic elements

are flowing around the world at an unprecedented speed and scale. Globalisation does not mean the inevitable elimination of national borders, but economic globalisation reflects the fact that the degree of interdependence in the international community has been strengthened and deepened. On this basis, the connotation and representation of national sovereignty are constantly being challenged.

Throughout the history of mankind, the sea has not only played the role of a nourisher providing food, but has also served as an interface for communication. Through this interface, people in different places have been able to fulfil their wishes for adventure, communication, interaction, trade, and even war. By way of using this interface, humans have gradually built up a legal system to regulate the use of the sea.

As discussed above, the international community's attitude towards marine living resources has evolved from "possession and use" to "conservation and management", and there are currently two main directions of thinking about the regulation of "conservation and management":

1. To protect marine living resources from extinction and to compile a reasonable and safe maximum catch of fish that can adequately supply human protein without wasteful overfishing through competitive fish catch records;

2. By means of these norms, coastal states or fishing nations can cooperate fully in complying with the normative agreement in a regional or sub-regional context, thereby reducing conflicts between them and promoting regional or sub-regional harmony.

This is achieved through the regulation of international trade rules, regional or sub-regional cooperation, and the management of related fishing practices. The emergence of this concept of "ocean governance" is a clear sign of the changing attitude of the international community toward oceans as a whole. "Rather than being a term for the management of the marine environment and resources, 'ocean governance' is a term for the reflection of mankind after a long period of use of the oceans, with the aim of ensuring that both ocean space and resources are used effectively and in a way that achieves sustainable resources.

From the above analysis of the development of the regulation of the international fisheries legal system, the development and influence of 'globalisation' plays an important role, whether from the perspective of environmental protection and resource conservation, from the impact of international trade practices, or from the regulation of international organisations.

Under the influence of the globalisation of concepts and through the development of international trade in fishery products, a broad platform for the promotion of sustainability has been established, which has led to a deepening concept of conservation and management of marine living resources and has, in turn, influenced changes in national legislation and policies.

At the same time, the collective forces of the international community are also pushing for the consolidation of such concepts. The RFMOs, for example, are attempting to construct legal norms that break away from the traditional international law framework of national sovereignty. Traditional international law principles such as "*pacta terrtiis nec nocent nec prosunt*", "freedom of fishing on the high seas" and "flag state control" are being challenged under the premise of conservation and management, and further developments are expected.

**Author Contributions:** The contribution for this research of the authors are as follows: K.-H.W.: Conceptualization; Formal analysis; Writing—original draft; H.-J.T.: Data curation; Supervision; Writing—review and editing. All authors have read and agreed to the published version of the manuscript.

**Funding:** This research was funded by National Science and Technology Council, Taiwan, Grant Number MOST 111-2425-H-110-001.

**Institutional Review Board Statement:** Not applicable.

**Informed Consent Statement:** Not applicable.

**Data Availability Statement:** Not applicable.

**Conflicts of Interest:** The authors declare no conflict of interest.

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
