# Peer review of "The Impact of Globalisation on the Development of International Fisheries Law"

_sustainability, doi:10.3390/su15075652_

Round 1
Reviewer 1 Report
Overall, this is an interesting paper that I enjoyed reading, but I would recommend that the authors consider a number of minor modifications/additions, as follows:
1. In section 3.1.4 I would recommend that the authors consider replacing ‘specific fish stocks’ with ‘straddling and highly migratory fish stocks’. The reason for this is that straddling and highly migratory fish stocks are border-crossing and therefore require international cooperation for management, whereby other fish species do not.
2. In section 3.2 on the trade of fish products, the authors may want to expand their analysis to other important instruments. As the authors mention regional agreements, perhaps they should also include the EU IUU Regulation (Council Regulation 1005/2008), which has had a significant impact on international fish trade. The financial clout of the EU market in fish product importation is mentioned at the start of the paper, so it would make sense to highlight this instrument (although the authors may not want to make an incursion into 'IUU fishing' mechanisms as such, though they are closely related to the theme of globalisation that they discuss). Another important instrument that is significant for international fish trade is the 2009 Port State Measures Agreement, introducing port mechanisms to restrict the introduction of IUU fishing products through fishing ports.
3. The authors make important statements in the conclusion, such as ‘The RFMOs, for example, are attempting to construct legal norms that break away from the traditional international law framework of national sovereignty. Traditional international law principles such as “pacta terrtiis nec nocent nec prosunt”, “freedom of fishing on the high seas” and “flag state control” are being challenged under the premise of conservation and management, and further developments are expected.’ A little more explanation is required on this point, as it is not entirely clear how the RFMOs do this - for legal scholars, a challenge to the pacta tertiis principle might seem to stem from Article 8(4) of the 1995 Fish Stocks Agreement in respect of RFMOs, but in reality there is no challenge because the provision only applies vis-a-vis other parties of the 1995 agreement - see Franckx (2000). Lastly, there seems to be part of a sentence missing from the text between footnotes 39 and 40.
Author Response
- Thanks to Reviewer’s suggestion, the term “specific fish stocks” in sub-topic has been replaced with “straddling and highly migratory fish stocks”.
- Thanks to Reviewer’s suggestion, however, the authors mentioned at the end of Section 1 Introduction that “In this paper, the authors present the findings by assessing changes in international fisheries law roughly during the two decades between the signing of UNCLOS and the completion of IPOA-IUU”. Therefore, the period for this study is limited between 1982 and 2001. Nonetheless, the authors would like to express the study clearer and include the Reviewer’s opinion that “Other important developments, such as the EU IUU Regulation (Council Regulation No. 1005/2008 of 29 September 2008) and Agreement on Port State Measures to Prevent, Deter and Eliminate Illegal, Unreported and Unregulated Fishing, PSMA (2009) will be dealt with in another research work” at the end of the Section 1.
- In terms of the discussion on the “pacta tertiis principle”, the authors’ arguments is that while international law generally deals with actors that are mostly states or international organizations. However, there are non-state actors in international activities today. In particular, "fishing entities" is one of the examples. For this, the authors suggest that this principle might be challenged.
- The text between footnotes 39 and 40 is revised as “In fact, Article 63, paragraph 2, of UNCLOS has foreseen the importance of establishing a cooperative mechanism for the conservation of fishery resources in high seas areas, that is to encourage cooperation through appropriate subregional or regional organizations. In addition, Article 64 adds an …”. Thanks to Reviewer’s comments.
Reviewer 2 Report
Review comments on Wang and Tsai
The Impact of Globalisation on the Development of International Fisheries Law
The aim of the study was to discuss the development of globalisation and its evolution of international law in relation to international fisheries law and its role in protection and conservation of high seas fisheries resources. To achieve this aim, the authors reviewed changes that occurred and the various conventions and reports that have resulted, at the various national and international forums that have been conducted over a 20-year period, and their influence on changes to international laws. They then present and analyse the results of this review and describe and discuss how the community attitudes to management of marine living resources have changed over this period.
Overall, the data presented in the paper achieves the study aims and provides an original contribution to improving our knowledge by reviewing how globalisation has influenced the development of international fisheries law. As such, I consider that a revised paper incorporating the marked comments on the attached PDF and addressing the comments detailed below could be resubmitted for assessment for publication.
Specific comments on the manuscript are detailed below.
No line numbers were included on the review manuscript. These should be included to assist masking review comments.
Throughout the manuscript the authors include numerous footnotes and references many of which could have been included in the reference list. The authors should consult the journals requirements regarding the use of footnotes and references.
The authors should fill in author information details on pages 15 and 16.
At a number of points in the manuscript Information is repeated in subsequent sentences. Examples of these are marked in the text. The authors should review the text to ensure that there is no repletion of information.
Given that the author’s first language is not English, I have marked on the attached PDF document, numerous suggested spelling and grammatical changes to improve the quality and flow of the manuscript. The authors should review these changes and have the manuscript reviewed before resubmission.
In particular, the number of spelling and grammatical errors in the manuscript indicates that the authors have not bothered to undertake a basic spelling and grammar check of the manuscript using a word processing program or had the document review before submitting to the journal. This review should be undertaken prior to resubmitting the manuscript.
Page 3 par 1. What does this text mean? Revise to clarify the text.
…..provide legitimacy for normative perspective to deal with global environmental problems.
Page 3 par 2 define what IPOA-IUU means.
Page 4 par 2. What does this text mean? Revise to clarify the text.
…on those issues which are even far way.
Page 4 par 3. What does this text mean? Revise to clarify the text.
Through the observations, it is understandable----
Page 6 par 2. What does this text mean? Revise to clarify the text.
The excessive growth of the economy and industry…..
Population, land use, agriculture,….
Page 6 par 2. This figure is based on a 1994 reference. Do the authors have a more recent reference for this issue.
It is estimated that by the year 2020,….
Page 6 par 3. What does this text mean? Revise to clarify the text.
In its Part XII, the UNCLOS provides pollution from land….
Page 10 par 5. What does this text mean? Revise to clarify the text.
Recent developments have therefore tended…..
Page 12 par 4. What does this text mean? Revise to clarify the text.
In general, in State practices,…..
Page 12 par 5. What does this text mean? Revise to clarify the text.
….but the impulsion of international non-governmental organisations……
Page 13 par 2. What does this text mean? Revise to clarify the text.
…..resolutions of international organisations with a normative nature.
There are several issues with the references with inconsistencies between format of different references. For some references, the first letter of each word in the paper title is capitalised and should be lower case. In other references, only the first word in the title is capitalised. The authors should check previous issues of the journal regarding presentation of references.

Author Response
- The authors would like to express our sincere gratitude to the Reviewer, especially for the detailed and patient comments, which have helped to improve the quality of the manuscript. The authors’ responses are as follows.
- Reviewer’s opinion: Page 3 par 1. What does this text mean? Revise to clarify the text. “…provide legitimacy for normative perspective to deal with global environmental problems.”
The sentence has been revised as “Although these declarations and conference reports are not legally binding, they still provide the legal framework for normative perspectives to deal with global environmental problems.”
- Reviewer’s opinion: Page 3 par 2 define what IPOA-IUU means.
The term “IPOA-IUU” was discussed in a latter section (Section 3.1.6), so a footnote was added to express the cross-reference with Section 3.1.6 and note 31.
- Reviewer’s opinion: Page 4 par 2. What does this text mean? Revise to clarify the text. “…on those issues which are even far way”.
The authors revised the sentence as: “Fourthly, globalisation will generate influence on those issues even if they have happened far away.”
- Reviewer’s opinion: Page 4 par 3. What does this text mean? Revise to clarify the text. Through the observations, it is understandable----
The authors deleted “Through the observations,” which will be more perceivable. Thanks.
- Reviewer’s opinion: Page 6 par 2. What does this text mean? Revise to clarify the text.
The excessive growth of the economy and industry…..
Population, land use, agriculture,….
The authors revised the text into “Rapid population growth, excessive land use, agriculture production, deforestation, fishery resources over-exploitation, urban development, and industrial emissions all affect the marine environment.”
- Reviewer’s opinion: Page 6 par 2. This figure is based on a 1994 reference. Do the authors have a more recent reference for this issue. “It is estimated that by the year 2020,….”
The authors revised the text with more recent reference, “According to the Sustainable Development Goals Report 2022, between 2009 and 2018, the world lost about 14% of coral reefs, often called the “rainforests of the sea” because of the extraordinary biodiversity they support. The oceans are also under increasing stress from multiple sources of pollution, which is harmful to marine life and eventually makes its way into the food chain. Among those sources of pollution, 80% comes from land-based activities. In 2021, a study estimated that more than 17 million metric tons of plastic entered the world’s ocean, making up about 85% of marine litter. The volume of plastic pollution entering the ocean each year is expected to double or triple by 2040, threatening all marine life. Moreover, in terms of fishery resources, global fish stocks are under increasing threat from overfishing and from IUU fishing. More than a third (35.4%) of global stocks were overfished in 2019, up from 34.2% in 2017 and 10% in 1974. In addition, the rapidly growing consumption of fish (an increase of 122% between 1990 and 2018), along with inadequate public policies for managing the sector, have led to depleting fish stocks.”
- Page 6 par 3. What does this text mean? Revise to clarify the text. In its Part XII, the UNCLOS provides pollution from land….
The authors revised the text, “Part XII of the UNCLOS stipulates that the sources of pollution from land, sea and atmosphere, as well as the consequences of pollution resulting from economic development activities that exploit marine resources. Obviously, over-fishing is one of the activities. However, this convention is just the beginning and there have been numerous meetings and documents on the regulation of high seas fishing activities.”
- Page 10 par 5. What does this text mean? Revise to clarify the text. Recent developments have therefore tended…..
The sentence is revised as “As a result, there is a tendency for governments or international organisations to discuss environmental protection issues in conjunction with trade activities and to link them more closely.”
- Page 12 par 4. What does this text mean? Revise to clarify the text. In general, in State practices,…..
The authors revised the text, “That is to say, State practices have shown that using eco-labelling to encourage and educate consumers to consume fish or fish products that have been processed from sources that are sustainable.”
- Page 12 par 5. What does this text mean? Revise to clarify the text. ….but the impulsion of international non-governmental organisations……
The authors revised the sentence as, “Whether the use and requirements for eco-labelling will constitute a non-tariff (or technical) barrier to trade on fish and fish products remains to be explored, but the impulsion of international non-governmental organisations is a demonstration of the impact of trade globalisation.”
- Page 13 par 2. What does this text mean? Revise to clarify the text. …..resolutions of international organisations with a normative nature.
The text is revised as, “For the practices of international society, international organisations are not only the product of international law but, in terms of the development content of legal norms, resolutions made by international organisations have also become one of the sources of international law. In addition, this kind of legislation will even bind States and other actors.”
- The authors would like to express our sincere gratitude to the Reviewer, especially for the detailed and patient comments, which have helped to improve the quality of the manuscript. The authors’ responses are as follows.
- Reviewer’s opinion: Page 3 par 1. What does this text mean? Revise to clarify the text. “…provide legitimacy for normative perspective to deal with global environmental problems.”
The sentence has been revised as “Although these declarations and conference reports are not legally binding, they still provide the legal framework for normative perspectives to deal with global environmental problems.”
- Reviewer’s opinion: Page 3 par 2 define what IPOA-IUU means.
The term “IPOA-IUU” was discussed in a latter section (Section 3.1.6), so a footnote was added to express the cross-reference with Section 3.1.6 and note 31.
- Reviewer’s opinion: Page 4 par 2. What does this text mean? Revise to clarify the text. “…on those issues which are even far way”.
The authors revised the sentence as: “Fourthly, globalisation will generate influence on those issues even if they have happened far away.”
- Reviewer’s opinion: Page 4 par 3. What does this text mean? Revise to clarify the text. Through the observations, it is understandable----
The authors deleted “Through the observations,” which will be more perceivable. Thanks.
- Reviewer’s opinion: Page 6 par 2. What does this text mean? Revise to clarify the text.
The excessive growth of the economy and industry…..
Population, land use, agriculture,….
The authors revised the text into “Rapid population growth, excessive land use, agriculture production, deforestation, fishery resources over-exploitation, urban development, and industrial emissions all affect the marine environment.”
- Reviewer’s opinion: Page 6 par 2. This figure is based on a 1994 reference. Do the authors have a more recent reference for this issue. “It is estimated that by the year 2020,….”
The authors revised the text with more recent reference, “According to the Sustainable Development Goals Report 2022, between 2009 and 2018, the world lost about 14% of coral reefs, often called the “rainforests of the sea” because of the extraordinary biodiversity they support. The oceans are also under increasing stress from multiple sources of pollution, which is harmful to marine life and eventually makes its way into the food chain. Among those sources of pollution, 80% comes from land-based activities. In 2021, a study estimated that more than 17 million metric tons of plastic entered the world’s ocean, making up about 85% of marine litter. The volume of plastic pollution entering the ocean each year is expected to double or triple by 2040, threatening all marine life. Moreover, in terms of fishery resources, global fish stocks are under increasing threat from overfishing and from IUU fishing. More than a third (35.4%) of global stocks were overfished in 2019, up from 34.2% in 2017 and 10% in 1974. In addition, the rapidly growing consumption of fish (an increase of 122% between 1990 and 2018), along with inadequate public policies for managing the sector, have led to depleting fish stocks.”
- Page 6 par 3. What does this text mean? Revise to clarify the text. In its Part XII, the UNCLOS provides pollution from land….
The authors revised the text, “Part XII of the UNCLOS stipulates that the sources of pollution from land, sea and atmosphere, as well as the consequences of pollution resulting from economic development activities that exploit marine resources. Obviously, over-fishing is one of the activities. However, this convention is just the beginning and there have been numerous meetings and documents on the regulation of high seas fishing activities.”
- Page 10 par 5. What does this text mean? Revise to clarify the text. Recent developments have therefore tended…..
The sentence is revised as “As a result, there is a tendency for governments or international organisations to discuss environmental protection issues in conjunction with trade activities and to link them more closely.”
- Page 12 par 4. What does this text mean? Revise to clarify the text. In general, in State practices,…..
The authors revised the text, “That is to say, State practices have shown that using eco-labelling to encourage and educate consumers to consume fish or fish products that have been processed from sources that are sustainable.”
- Page 12 par 5. What does this text mean? Revise to clarify the text. ….but the impulsion of international non-governmental organisations……
The authors revised the sentence as, “Whether the use and requirements for eco-labelling will constitute a non-tariff (or technical) barrier to trade on fish and fish products remains to be explored, but the impulsion of international non-governmental organisations is a demonstration of the impact of trade globalisation.”
- Page 13 par 2. What does this text mean? Revise to clarify the text. …..resolutions of international organisations with a normative nature.
The text is revised as, “For the practices of international society, international organisations are not only the product of international law but, in terms of the development content of legal norms, resolutions made by international organisations have also become one of the sources of international law. In addition, this kind of legislation will even bind States and other actors.”
Reviewer 3 Report
The paper is well writen and a possible good reference for fishereis management. Some comments that need to be incorporated in the paper are written as follows:
1. In the abstract starting on statement of “The purpose of this paper is to discuss …. Till to the end of the abstract” should be written in the Past Tenses form NOT in the Present Tenses.
2. 2. In Intro, authors describe long about globalization yet very little (none) description regarding globalization in international fisheries laws. It would be better to describe a brief history of some international laws that affect globalization in fisheries laws.
3. 3. It seems that the paper cited the regulations before the year of 2000. Yet there were many new regulations or world declarations regarding fisheries management after the year of 2000. It is very important to cited the newest worldwide regulation or world declaration on fisheries management for this paper.
4. 4. It is very important to describe how the world regulations or declarations can affect the regional or countries fisheries management.
Author Response
- The tense of the abstract had been changed to PAST ones. Thanks to the Reviewer’s comments.
- Several sentences have been inserted into the last paragraph of Section 1 (Introduction).
This is particularly evident in the high seas fisheries, where some conditions have emerged for what was originally a matter of freedom of fishing on the high seas. For example, there are: the competence of coastal States to manage fisheries resources, and the limits to which attention should be paid in catching highly migratory fish stocks and straddling fish stocks.
- Thanks to the Reviewer’s suggestion on adding those international instruments concerning the development of international fisheries law which was adopted after 2000. However, the authors mentioned at the end of Section 1 Introduction that “In this paper, the authors present the findings by assessing changes in international fisheries law roughly during the two decades between the signing of UNCLOS and the completion of IPOA-IUU”. Therefore, the period for this study is limited to the period between 1982 and 2001. The authors are conducting another research which pays attention to the development of international fisheries law after 2000, such as “the EU IUU Regulation (Council Regulation No. 1005/2008 of 29 September 2008)” and “Agreement on Port State Measures to Prevent, Deter and Eliminate Illegal, Unreported and Unregulated Fishing, PSMA (2009)” and others. These instruments will be dealt with in another research work.
- Thanks to the Reviewer’s comments on encouraging to describe how international instruments affect fisheries management measures. The authors mentioned some challenges generated by the development of the international fisheries law, which includes, the competence of international organisations, principle of pacta terrtiis nec nocent nec prosunt, freedom of fishing on the high seas, and flag state control. These issues concern the interaction and interrelations between international instruments and States policies.
Round 2
Reviewer 2 Report
No issues identified in the revised manuscript.